# Pharmacokinetic Study of Islatravir and Etonogestrel Implants in Macaques

**DOI:** 10.3390/pharmaceutics15122676

**Published:** 2023-11-26

**Authors:** Michele B. Daly, Andres Wong-Sam, Linying Li, Archana Krovi, Gregory J. Gatto, Chasity Norton, Ellen H. Luecke, Victoria Mrotz, Catalina Forero, Mackenzie L. Cottrell, Amanda P. Schauer, Joy Gary, Josilene Nascimento-Seixas, James Mitchell, Ariane van der Straten, Walid Heneine, J. Gerardo Garcίa-Lerma, Charles W. Dobard, Leah M. Johnson

**Affiliations:** 1Division of HIV/AIDS Prevention, National Center for HIV/AIDS, Viral Hepatitis, STD, and TB Prevention, Centers for Disease Control and Prevention, Atlanta, GA 30329, USA; 2RTI International, Durham, NC 27709, USA; 3Division of Scientific Resources, National Center for Emerging and Zoonotic Infectious Diseases, Centers for Disease Control and Prevention, Atlanta, GA 30329, USA; 4Eshelman School of Pharmacy, University of North Carolina, Chapel Hill, NC 27599, USA; 5ASTRA Consulting, Kensington, CA 94708, USA; 6Center for AIDS Prevention Studies, Department of Medicine, University of California San Francisco, San Francisco, CA 94104, USA

**Keywords:** HIV pre-exposure prophylaxis, islatravir, biodegradable implant, multipurpose prevention technologies, etonogestrel

## Abstract

The prevention of HIV and unintended pregnancies is a public health priority. Multi-purpose prevention technologies capable of long-acting HIV and pregnancy prevention are desirable for women. Here, we utilized a preclinical macaque model to evaluate the pharmacokinetics of biodegradable ε-polycaprolactone implants delivering the antiretroviral islatravir (ISL) and the contraceptive etonogestrel (ENG). Three implants were tested: ISL-62 mg, ISL-98 mg, and ENG-33 mg. Animals received one or two ISL-eluting implants, with doses of 42, 66, or 108 µg of ISL/day with or without an additional ENG-33 mg implant (31 µg/day). Drug release increased linearly with dose with median [range] plasma ISL levels of 1.3 [1.0–2.5], 1.9 [1.2–6.3] and 2.8 [2.3–11.6], respectively. The ISL-62 and 98 mg implants demonstrated stable drug release over three months with ISL-triphosphate (ISL-TP) concentr54ations in PBMCs above levels predicted to be efficacious for PrEP. Similarly, ENG implants demonstrated sustained drug release with median [range] plasma ENG levels of 495 [229–1110] pg/mL, which suppressed progesterone within two weeks and showed no evidence of altering ISL pharmacokinetics. Two of the six ISL-98 mg implants broke during the study and induced implant-site reactions, whereas no reactions were observed with intact implants. We show that ISL and ENG biodegradable implants are safe and yield sufficient drug levels to achieve prevention targets. The evaluation of optimized implants with increased mechanical robustness is underway for improved durability and vaginal efficacy in a SHIV challenge model.

## 1. Introduction

Globally, there is a growing recognition of HIV-associated gender disparities, with young women having increased rates of new HIV infections and higher rates of HIV-related mortality than men [1,2]. Pre-exposure prophylaxis (PrEP) is highly effective in preventing HIV; however, PrEP products must be desirable to the user, a quality that can vary by age, gender, and location, among others [3,4]. Women of reproductive age often seek medical care for pregnancy prevention, and offering PrEP in tandem with a contraceptive, as a multi-purpose prevention technology (MPT), has great potential to address reproductive health concerns for women while increasing PrEP use, thus mitigating HIV-related gender disparities [5,6]. 

Daily oral regimens are highly effective for both birth control and HIV PrEP if taken consistently. However, persistence to these oral medications is difficult for many users, and missed or late doses can reduce the contraceptive or HIV prevention effectiveness [7,8]. There are multiple long-acting contraceptive options, such as intrauterine devices, subdermal implants, intravaginal rings, and long-acting injectables, available to women who prefer these methods over daily oral medication or barrier methods [9]. Nexplanon^®^, a subdermal implant that releases the hormone etonogestrel (ENG) for three years, is the only hormonal contraceptive implant that is approved by the U.S. Food and Drug Administration (FDA) [10]. For long-acting PrEP, there are currently two options: Apretude^®^, a long-acting formulation of cabotegravir that is administered intramuscularly every 2 months [11], and the monthly intravaginal Dapivirine ring, approved in several African countries to prevent acquisition from vaginal sex [12]. Furthermore, there are many long-acting HIV PrEP products in the clinical pipeline, including subdermal implants, intravaginal rings, injectables, infusions, microarray patches, and oral regimens with reduced dosing frequency [13]. 

Implants are capable of ultra-long-acting drug release and, unlike injectables, can be removed if safety issues arise or user preferences change. A range of PrEP implants have been evaluated in early clinical or preclinical models including non-biodegradable implants (ethylene co-vinyl acetate [14,15,16], silicone [17,18], polyurethane [19], and titanium [20]) and implants constructed using biodegradable polymers (ε-polycaprolactone (PCL) [14,21,22] or polylactic acid [14]). Several of these implant technologies have been engineered to release the antiretroviral tenofovir alafenamide (TAF) and tested in preclinical models. Biodegradable PCL implants showed stable release of TAF (0.7 mg/day) and complete protection against vaginal SHIV infection in macaques; however, local inflammation and tissue necrosis developed at the implantation site after 8 weeks, a reaction that was not observed with the placebo PCL implants [23,24]. Implant-site necrosis was also observed with other TAF implant platforms [25,26,27], although some only induced a mild inflammatory response [26]. These TAF-related adverse reactions encouraged implant developers to pivot and investigate other antiretrovirals for long-term sustained release.

Islatravir (ISL) is a first-in-class nucleoside reverse transcriptase translocation inhibitor (NRTTI) that is phosphorylated within cells to ISL-triphosphate (ISL-TP), the active metabolite that inhibits viral replication [28]. The high potency and long half-life of ISL prompted the evaluation of multiple dosing strategies for HIV prevention or treatment in humans including monthly doses of ISL and sustained ISL release from nonerodable implants [15,16,29]. However, in 2021, the FDA placed clinical holds on all ISL trials after lymphopenia was observed across clinical studies [30]. Subsequent analyses indicated that ISL-induced lymphopenia was dose-exposure dependent and reversible after treatment discontinuation, but the exact mechanism of toxicity remains unknown [31]. ISL-TP concentrations and the corresponding changes in lymphocyte counts were used to model an optimized ISL dose that would be safe and effective [32]. Clinical trials are moving forward with an oral dose of 0.25 mg once daily (NCT05705349, NCT05631093, NCT05630755) or 2 mg once weekly (NCT05052996) for HIV treatment. The status of NCT05115838, which will evaluate ISL implants (47–57 mg) for HIV PrEP, is listed as ongoing but not yet recruiting. It is thus important to continue evaluating ISL doses capable of providing safe and effective PrEP.

In this pilot study, we evaluated the safety and pharmacokinetics of biodegradable PCL implants releasing ISL or ENG in a preclinical macaque model. The primary objectives were to monitor for adverse implant-site reactions and select implant candidates with desired pharmacokinetics for future PrEP efficacy studies. Three implants (ISL-62 mg, ISL-98 mg, and ENG-33 mg) were evaluated in normal cycling female pigtailed macaques. When co-implanted, ISL-62 mg and ISL-98 mg implants released ISL at levels that surpassed the predicted PrEP efficacy threshold for ISL-TP levels in peripheral blood mononuclear cells (PBMCs) (50 fmol/10^6^ cells) [33]. ENG-33 mg implants inhibited progesterone production, indicating that the ENG levels were sufficient for suppressing ovulation in macaques. Two of the six ISL-98 mg implants broke during the study, resulting in swelling and redness at the implantation sites. However, intact ISL and ENG implants did not induce implant-site reactions, indicating that the reactions observed at the broken implant sites were likely due to sudden high drug exposure after rod breakage. Our data support the continued development of biodegradable ISL and ENG implants as an MPT for HIV PrEP and contraception.

## 2. Materials and Methods

### 2.1. Implant Fabrication and Sterilization

ISL, ENG, and pharmaceutical-grade sesame oil were purchased from WuXi AppTec (Shanghai, China), AdooQ^®^ Bioscience (Irvine, CA, USA), and Croda (Snaith, UK), respectively. Medical-grade PCL (PC-17) pellets were purchased from Corbion (Amsterdam, The Netherlands). Hollow PCL tubes (2.5 mm outer diameter (O.D.) with a wall thickness of 100 µm or 300 µm to achieve targeted release rates) were manufactured via a hot-melt, single extrusion process using solid PC-17 pellets at GenX Medical (Chattanooga, TN, USA). The dimensions of the extruded tubes were verified with a 3-axis laser measurement system and light microscopy at GenX Medical. 

PC-17 tubes were trimmed to the desired length (16 mm, 25 mm, or 40 mm) with 3 mm of headspace at each end for sealing and were enclosed at one end using custom-built sealing apparatus, as previously described [21,22]. ISL was mixed with sesame oil (1:1 *w/w* ratio) and loaded into the implant core using a 3 mL syringe fitted with a 14-gauge blunt-tip stainless steel needle. ENG was mixed with sesame oil (2:1 *w*/*w* ratio) and loaded into the implant core using a stainless-steel rod. Once the implants were loaded with a sufficient quantity of formulation that achieved the desired reservoir length, the other end of the implant was sealed. Implants were then subjected to gamma irradiation (18–24 kGy) on a continuous path for 8 h using a Cobalt-60 gamma ray source (Nordion Inc., Ottawa, ON, Canada) at STERIS Applied Sterilization Technologies (Libertyville, IL, USA).

### 2.2. Determining Release Rates (In Vitro) and Drug Purity (In Vitro and In Vivo)

To estimate the in vitro release rate, we followed previously validated methods [22]. Briefly, each implant (*n* = 3 per group) was submerged in an individual sterile polypropylene tube filled with 1× phosphate-buffered saline (PBS, pH 7.4) and incubated at 37 °C with gentle agitation (104 rpm). Implants were transferred (2×/week) to fresh buffer solutions to maintain sink conditions for the duration of the study. The concentrations of ISL or ENG in buffer release media were measured with high-performance liquid chromatography (HPLC) using an Agilent 1100/1200 HPLC-UV system equipped with an Agilent Zorbax Bonus-RP, 4.6 × 150 mm, 3.5 µm column. Samples were analyzed using an injection volume of 5 µL and a flow rate of 0.8 mL/min, with 0.01% TFA as Mobile Phase A and acetonitrile as Mobile Phase B. A run time of 30 min was used, along with a diode array detector (DAD) of 220 nm for ENG and a DAD of 260 nm for ISL. 

At the terminal time points, the drug formulation was extracted from each implant (in vitro and in vivo) via sonication and analyzed for purity with HPLC using ISL and ENG standards purchased from WuXi AppTec (Shanghai, China) and AdooQ^®^ Bioscience (Irvine, CA, USA), respectively. The excised implants from macaques were gamma sterilized prior to formulation extraction.

### 2.3. Animal Care Guidelines

This research was conducted under a Centers for Disease Control and Prevention (CDC) Institutional Animal Care and Use Committee (IACUC)-approved protocol in compliance with the Animal Welfare Act, PHS Policy, and other Federal statutes and regulations relating to the use of animals in research. Animals were housed in an AAALAC International accredited facility that adheres to principles stated in the ‘Guide for the Care and Use of Laboratory Animals’ [34]. All macaques were deemed healthy based on physical examination and baseline bloodwork. Every effort was made to minimize distress through enrichment opportunities and humane interactions. While assigned to the study, the macaques received food twice daily (LabDiet 5045 and 5049. LabDiet, St Louis, MO, USA) and water ad libitum via an automatic drinking valve. Wellness checks were conducted twice daily, and animals were given enrichment in the form of toys and food. Macaques were housed in ABSL2 rooms at 17.8–28.9 °C with a 30–70% relative humidity and 12:12 h light/dark cycle, and socially housed whenever possible. Prior to implantation, pair-housed and single-housed animals had cage dividers that, at minimum, allowed eye contact. After implantation, animals were placed in protected contact housing to minimize the potential for implant injury, where they were still able to see each other and have minimal contact.

All animal procedures were performed under anesthesia. The following sedation protocols were utilized throughout the study, chosen based on anesthetic needs and procedure time: ketamine (100 mg/mL; Dechra Veterinary Products, Overland Park, KS, USA) was administered at a dose of 10 mg/kg; dexmedetomidine (0.5 mg/mL Dexmedesed, Dechra Veterinary Products, Overland Park, KS, USA) was administered at a dose range of 0.01–0.013 mg/kg in combination with ketamine at a dose range of 3–5 mg/kg; atipamezole (Antisedan, 5.0 mg/mL, Zoetis, Parsippany, NJ, USA) was administered at a dose of 0.3 mg/kg to reverse the effects of dexmedetomidine; telazol (Tiletamine–zolazepam, Zoetis, Parsippany-Troy Hills, NJ, USA) at a dose range of 2–6 mg/kg. Drugs were administered intramuscularly (IM) using squeeze-back cages. Anesthetic events were conducted in ABSL2.

### 2.4. Implantation and Removal

Once anesthetized, the macaques were placed in dorsal/posterior recumbency with both arms extended. The implantation site (the medial aspect of the biceps brachii muscle) was clipped of hair and aseptically prepped with a triplicate application of Betadine solution (Povidone-Iodine 5%, Purdue Frederick Company, Stamford, CT, USA) and sterile saline. Clear plastic sterile drapes were placed over the site to maintain sterility during implantation. A small incision was made along the medial bicipital groove with a #10 scalpel. The implant was inserted subcutaneously using a 2.7 mm trocar kit (Shinva Ande Healthcare Apparatus Co., Ltd., Zibo City, China). Implants were removed in a similar manner, first clipping hair, aseptically preparing the site, and then making a small scalpel incision. The skin over the implant was grasped and the implant was pressed toward the incision site until visible. The tissue capsule was carefully snipped, allowing the implant to easily be pushed out. The incision was closed with GLUture^®^ topical tissue adhesive (Formulated Cyanoacrylate, Abbott Animal Health, Abbott Park, IL, USA) after each procedure. After the implantation and removal procedures, Meloxicam SR (10 mg/mL, ZooPharm, Windsor, CO, USA) and Buprenorphine SR (10 mg/mL, ZooPharm, Windsor, CO, USA) were administered subcutaneously as a single dose of 0.6 mg/kg and 0.2 mg/kg, respectively. 

### 2.5. Pharmacokinetic Study Design

The pharmacokinetic profile of implants releasing ISL with or without ENG was investigated in 6 female pigtailed macaques (mean age, 13 years [range 7–15]; mean weight 7.95 kg [range 7.10–11.45]). Using a staggered implantation design, three ISL doses were evaluated: low dose (one ISL-62 mg implant), mid dose (one ISL-98 mg implant), and high dose (one ISL-62 + one ISL-98 mg implant). ENG-33 mg implants were evaluated concurrently in the mid-ISL study group. Group 1 consisted of three SHIV+ macaques that were implanted with an ISL-62 mg implant in the right arm and monitored for 36 days to assess the low ISL dose. The same group of macaques received an ISL-98 mg implant in the left arm at day 36, receiving a high dose of ISL. Implanted animals were monitored for a total of 113–135 days prior to euthanasia. The second group of three macaques was implanted with an ISL-98 mg implant in the right arm (mid ISL dose), and 36 days later received an ENG-33 mg implant in their left arm to assess whether co-implantation altered the ISL pharmacokinetics. After 55 days of co-implantation, ISL implants were removed to determine the ISL PK tail. ENG implants were removed 198 days after implantation.

### 2.6. Blood, Tissue, and Swab Processing

Blood was collected in BD Vacutainer^®^ CPT™ Cell Preparation Tubes and PBMCs were isolated following the manufacturer’s protocol. Plasma aliquots of 500 µL were immediately frozen at −70 °C for ISL, ENG and progesterone analyses. PBMCs were treated with BioLegend red blood cell (RBC) lysis buffer (BioLegend, San Diego, CA, USA) and then counted using trypan blue exclusion on the Countess II FL (Invitrogen, Carlsbad, CA, USA). Vaginal and rectal mucosal fluids and tissue biopsies were collected once every two weeks. Swabs (Fisherbrand™ Synthetic-tipped Applicator, Waltham, MA, USA) were inserted in the vagina and rectum for five minutes to ensure uptake of mucosal secretions. The volume of fluid for each sample was determined by weighing the swabs pre- and post-collection. Tissue samples were taken with medical forceps (Boston Scientific Radial Jaw 4 Forceps, 2.2 mm jaw, 160 length), and three pinches were attempted at each site. Extraneous liquid was carefully removed, and tissue was weighed on an analytical balance. 

### 2.7. Measurement of Intracellular ISL-TP and dATP from PBMCs and Tissue

Following an established protocol, 5 million live PBMCs were resuspended in 500 µL ice-cold 80% methanol, vortexed for 1 min and immediately frozen at −70 °C [35]. ISL-TP and dATP were quantified using previously published LC-MS/MS methods [36]. The calibrated ranges were 0.05–200 and 0.005–100 ng/mL for ISL-TP and dATP, respectively. Accuracy and precision were within 20%. Concentrations were normalized to live cells and reported in fmol/million cells. Tissue biopsies were transferred to 1.5 mL tubes prefilled with ceramic beads (Omni International, Kennesaw, GA, USA), and 800 µL 80% MeOH was added. Tissues underwent bead mill homogenization using the Omni Bead Ruptor 24 and were then frozen at −70 °C. ISL, ISL-TP and dATP were quantified using LC-MS/MS. Briefly, analytes were extracted using protein precipitation with isotopically labeled internal standards (^13^C_15_, N_3_-EFdA (Wuxi AppTec, Shanghai, China); ^13^C_10_, ^15^N_5_-dGTP (Sigma-Aldrich, St. Louis, MO, USA), and ^13^C_10_; ^15^N_5_-dATP (Sigma-Aldrich, St. Louis, MO, USA) for ISL; and ISL-TP and dATP, respectively) then dried under nitrogen gas and reconstituted with 125 μL of 1 mM ammonium phosphate. Analytes were separated on an XTERRA MS C18 (40 × 2.1 mm 3.5 μm) column and then detected on an AB Sciex API-5000 triple quadrupole mass spectrometer under positive ionization mode. The calibrated range was 0.025–50 ng/mL. Accuracy and precision were within 20%. Concentrations were normalized to tissue mass and reported in ng/g (ISL) or fmol/g (ISL-TP and dATP) units. 

### 2.8. Measurement of ISL and ENG in Plasma and Mucosal Fluids

Plasma ISL and ENG were quantified by adapting previously published LC-MS/MS rat plasma methods to NHP plasma [22]. The calibrated ranges were 0.1–100 and 0.2–200 ng/mL for ISL and ENG, respectively. Accuracy and precision were within 15% with 20% at the lower limit of quantification (LLOQ). 

Mucosal fluids were eluted off of the swabs in 1 mL of 80:20 methanol:water and a 500 µL aliquot of the elution fluid was sent for ISL quantification by LC-MS/MS. Briefly, ISL was extracted using protein precipitation with isotopically labeled internal standard (^13^C_15_, N_3_-EFdA (Wuxi AppTec, Shanghai, China)) or ENG (Toronto Research Chemicals, North York, ON, Canada) and then dried under nitrogen gas and reconstituted in 75 μL of purified water. ISL was separated on a Waters Atlantis T3 (50 × 2.1 mm 3 μm) column and then detected on an AB Sciex API-5000 triple quadrupole mass spectrometer under positive ionization mode. The calibrated range was 0.02–100 ng/mL. Accuracy and precision were within 30%.

### 2.9. Measurement of Progesterone in Plasma

Plasma progesterone was measured by Wisconsin National Primate Center Assay Services. Briefly, plasma, standards and quality control samples were diluted in 500 μL of ultrapurified water (Fisher Scientific, Waltham, MA, USA). Methyl tert butyl ether (Fisher Scientific) was then added, vortexed vigorously, and incubated at room temperature for 5 min. The organic phase containing steroids was transferred into a new tube, evaporated to dryness using an air stream and heated water bath (60 °C), and then resuspended in 50 μL of 20% acetonitrile in water and analyzed using LC-MS/MS. The calibration curve ranged from 0.195 to 25 ng/mL and the linearity was r > 0.9990. The intraassay CV was 4.24% and the interassay CV was 6.47%. 

### 2.10. Implant-Site Reactions

Implantation sites were monitored weekly for adverse reactions including but not limited to erythema, edema, and necrosis. A modified Draize scale was used to grade erythema and edema of implant-site reactions (0, no reaction–4, severe), either in person or with high-resolution digital photography [23,37]. Draize scores from two veterinarians were averaged. After implant removal, two skin biopsies (4 mm, Integra, Princeton, NJ, USA) were collected at the medial or distal end of the implant site, H&E stained, and scored semiquantitatively by two pathologists using a lightly modified version of the international standard for biological evaluation of medical devices (ISO-10993-6 [38], severity scores for each parameter ranging from 0 (not present) to 5 (severe)).

### 2.11. Statistical Analysis

Simple linear regression was performed, and all figures were created using GraphPad Prism version 9.1.0 for Windows, GraphPad Software, San Diego, CA, USA.

## 3. Results

### 3.1. Implant Characteristics and Macaque Study Design

We evaluated two ISL implants (ISL-62 mg and ISL-98 mg) and one ENG implant (ENG-33 mg), which are described in Table 1. The ISL and ENG implants had wall thicknesses of 100 µm and 300 µm, respectively. The two ISL implants differed in the amount of drug loaded, length, and in vitro release rates. Figure 1 shows the in vitro drug release profile of the ISL and ENG implants. 

For the in vivo macaque study, we used a staggered implantation design to achieve three fixed doses of ISL based on in vitro release rates: low dose (one ISL-62 mg implant, in vitro release rate of 41.9 µg/day), mid dose (one ISL-98 mg implant, in vitro release rate of 65.6 µg/day), and high dose (one ISL-62 and one ISL-98 mg implants for a total cumulative in vitro release rate of 107.5 µg/day). For the low-dose group, 3 macaques had an ISL-62 mg implant inserted in their right arm and approximately one month later received an ISL-98 mg implant in their left arm, thus transitioning to the high-dose group (Figure 2A). Both implants were removed at euthanasia, approximately 3–4 months following the insertion of the ISL-98 and ISL-62 implants, respectively. A second group of three animals received an ISL-98 mg implant in the right arm and then received an ENG-33 mg implant in the left arm approximately one month later. The ISL-98 implant was removed ~3 months after implantation and the ENG-33 mg implant was removed ~6 months after implantation (Figure 2B). 

### 3.2. Pharmacokinetic Profile of ISL and ISL-TP in Blood

The concentrations of ISL in plasma and ISL-TP in PBMCs were measured as a function of time post-implantation (Figure 3A,B). Both analytes showed stable and sustained drug release, with the high-dose ISL group (ISL-62 mg and ISL-98 mg) achieving the PrEP target of 50 fmol ISL-TP/10^6^ PBMCs (Figure 3B and Table 2). The levels of plasma ISL and PBMC ISL-TP exhibited dose linearity and correlated with the in vitro release rate (Figure 3C). To better visualize the PK differences between the three dose levels, Figure 3A,B show all dosage groups on the same axis. For the low-dose (ISL-62 mg; blue) and mid-dose (ISL-98 mg; red) groups, day zero started at the time of implantation. However, for the high-dose (ISL-62 and ISL-98 mg; green) group, day zero started the day of the second implantation of the ISL-98 mg implant.

Two ISL-98 mg implants, one from the mid-dose ISL group and one from the high-dose ISL group, were deemed broken during the implant-site assessment on day 49 and 56 post-implantation, respectively (Appendix A). This observation coincided with a rapid increase in plasma ISL and PBMC ISL-TP. Due to the breakage, the amount of ISL being released was unknown; therefore, these data were excluded from the comprehensive pharmacokinetic analyses but shown individually (Appendix A). Overall, intact implants demonstrated stable release of ISL over the 3–4 months of study.

To evaluate the pharmacokinetic tail of ISL after implant removal, the ISL-98 mg implant was surgically removed from the mid-dose group at day 91. Plasma ISL and PBMC ISL-TP were undetectable within 7–14 days after removal (Figure 3D). 

### 3.3. ISL Penetration in Rectal and Vaginal Tissues

ISL-TP was detected in tissue homogenates from both vaginal and rectal compartments at all dose levels (Appendix A). In vaginal tissues, average ISL-TP concentrations at day 21 achieved with the low, mid, and high ISL doses were 9.6 (range, 8.5–11.6), 13.5 (range, 8.4–21.6), and 30.5 (range, 10.7–64.1) pmol/g of tissue, respectively (Table 2). Average ISL-TP levels in rectal tissues at day 21 were higher than in vaginal tissues; 18.5 (range, 8.6–27.3) pmol/g for the low dose, 20.5 (range, 12.5–34.0) pmol/g for the mid dose, and 36.9 (range, 26.0–45.4) pmol/g of tissue for the high dose (Table 2). ISL was also consistently detected in rectal and vaginal fluids (Appendix A). 

### 3.4. Suppression of Progesterone Production with ENG Implants

The mid-dose ISL macaques (*n* = 3) received one ENG-33 mg implant and were monitored for progesterone production and ENG levels in plasma (Figure 4). Plasma ENG concentrations reached C_max_ within 1–3 weeks (875 pg/mL [range 660–1110]) post-implantation and remained above 229 pg/mL for 198 days (Table 2). Prior to ENG-33 mg implantation, all animals had fluctuating progesterone concentrations indicative of the luteal and follicular phase of the menstrual cycle. After ENG-33 mg implantation, progesterone levels were suppressed in all animals below 0.3 ng/mL within 2 weeks. Plasma progesterone was suppressed during the entire ENG-33 mg implantation period, demonstrating the ability of ENG-eluting implants to effectively inhibit ovulation over several months. After ENG-33 mg implant removal, ENG was undetectable in plasma within 5 days and plasma progesterone levels began to rebound within 20 days (Figure 4).

### 3.5. Effect of ISL and ENG Co-Implantation on ISL Pharmacokinetics

To assess how co-implantation of ISL and ENG may alter pharmacokinetics, we compared ISL levels in blood and tissues from ISL-98 mg implanted macaques prior to and after ENG-33 mg implantations. The median plasma ISL levels were 2.1 nM (range, 1.2–2.8) and 1.8 nM (range, 1.6–6.3) prior to and after ENG-33 mg insertion, respectively (Figure 5A). The median ISL-TP levels in PBMCs two weeks after ISL-98 mg implantation were 33.6 fmol/10^6^ PBMCs (range, 21.2–61.2) and, after ENG-33 mg implantation, they were 48.8 (range, 22.5–143.0) (Figure 5B). In vaginal tissues, the ISL-TP levels were similar with or without ENG. In rectal tissues, ISL-TP was lower in the presence of ENG; however, this data set was limited as ISL-TP was detectable in rectal tissues at only one timepoint prior to ENG implantation (Figure 5C and Appendix A). 

ISL-TP competes with the natural nucleotide dATP for incorporation by reverse transcriptase, and some studies suggest that hormonal contraceptives can alter nucleotide metabolism [39,40,41]. To determine whether ENG affected nucleotide metabolism and possibly competition between ISL-TP and dATP, we compared ratios between ISL-TP and dATP among ISL-98 mg-implanted animals before and after ENG-33 mg implantation. A higher ratio indicates that more ISL-TP is present, increasing the likelihood of incorporation by reverse transcriptase and successful viral inhibition. Figure 5D shows that the ISL-TP:dATP ratios were unaffected in PBMCs and rectal tissues but increased in vaginal tissues in the presence of ENG. These data indicate that ENG does not reduce the active pharmaceutic agent, ISL-TP, in blood or tissues, and will likely have no adverse impact on viral inhibition as a result of enhanced dATP levels.

### 3.6. Implant Site Reactions

Implant sites were scored for erythema and edema using a modified Draize scale: 0 (no reaction) to 4 (severe) (Figure 6A). The median [range] Draize score was 0.0 [0.0–0.8] for the ENG-33 mg, 0.0 [0.0–1.5] for the ISL-62 mg and 0.3 [0.0–1.3] for the ISL-98 mg implants, excluding scores after implant breakage (Figure 6B). The two ISL-98 implants that broke on day 49 and 56 after implantation resulted in mild to well-defined redness and swelling, with a median Draize score (range) of 1.25 (range 0.8–2.0). There was no obvious cause of the break, e.g., scratching or picking at the implant site. Overall, intact ISL implants (*n* = 7/9) and ENG implants (*n* = 3/3) were safe and well-tolerated in macaques. 

Skin biopsies were collected immediately after ISL-98 mg implant removal at two sites, one central to the implant (medial) and one at the end (distal). Biopsies were H&E stained and semi-quantitatively scored using a slightly modified version of the international standard for biomedical device evaluation (ISO-10993-6 [38], severity scores for each parameter ranging from 0 (not present) to 5 (severe)). H&E staining showed that intact implants were pathologically unremarkable with a median score of 1 (range 0–55, adding the severity scores across all evaluated histologic parameters) (Table 3). A biopsy taken directly at the breakage site of the ISL-98 mg implant showed edema, early fibrosis, and mild necrosis/inflammation, receiving a score of 27 out of 55 (PT-5, medial). Interestingly, the biopsy taken distal to the breakage site was pathologically unremarkable with a score of 1, suggesting that the pathology was limited to the point of the breakage.

### 3.7. Purity of Drug and Predicted Duration of Release

After implant removal, the residual drug was assessed for purity (Table 1), showing >97% drug purity for ISL-62 mg and ISL-98 mg and >99% drug purity for ENG-33. Considering the in vitro release rates, the ISL-62, ISL-98, and ENG-33 implants have an expected duration of 4.1, 4.1, and 2.9 years, respectively. The estimated durations assume zero-order release kinetics and do not account for potential changes in dosing over time or tailing of the release profiles.

## 4. Discussion

MPTs for HIV PrEP and contraception can offer women combined protection in a single product [42]. Subdermal implants capable of long-term drug release and preventative efficacy are intriguing as MPTs as they can reduce the burden of daily oral medications and lessen clinic visits required for monthly ring replacement or injectable contraception or PrEP. Moreover, implants comprised of biodegradable polymers can advantageously bypass the need for removal, thus eliminating a minor surgical procedure to retrieve spent implants. A subdermal biodegradable PCL implant releasing tenofovir alafenamide (TAF) recently showed extended drug release and protective efficacy in a macaque SHIV challenge model, although further clinical advancement was halted due to drug-related implant-site reactions [23,24,27]. Here, we report on the preclinical assessment of next-generation PCL implants releasing the antiretroviral ISL, or the hormonal contraceptive ENG. We document prolonged release of ISL from two implant configurations that varied in length, drug load, and release rate, and demonstrated sustained release of ENG at clinically relevant levels. We also observed little to no implant site reactions when implants kept their integrity. Additional studies are needed to address the release kinetics of ISL and ENG co-formulated in a single implant.

Pharmacokinetic analysis in macaques of ISL implants showed that ISL analytes increased linearly with dose as estimated by the in vitro release rate. The data also revealed a strong correlation between ISL levels in plasma and ISL-TP concentrations in PBMCs, suggesting that plasma ISL may serve as a good surrogate for estimating ISL-TP levels in HIV target cells. Macaques that received two ISL implants, ISL-62 and ISL-98, had ISL-TP concentrations above the predicted protective PrEP benchmark of 50 fmols/10^6^ PBMCs [33]. In addition, ISL-TP was also detected in vaginal and rectal tissue, both important sites for sexual HIV transmission. Notably, the tissue ISL-TP levels in macaques (10–30 pmols/g) were slightly higher than those observed in the small number of human tissue samples [16,43]. The ISL levels in plasma were between 1.3 and 2.8 nM, which are within the range predicted to be safe and moving forward in clinical trials [32,44]. Compared with human studies, including the nonerodable ISL implant, we observed lower ISL-TP concentrations at equivalent plasma ISL values [15,16,44,45]. However, we calculated the ISL-TP:ISL ratios in other macaque studies and found a similar trend of lower ISL-TP in macaques compared to humans [14,36,46,47]. This could be attributed to differential rates of phosphorylation between species or possibly the sample processing method [36]. 

Recently, a refillable ISL implant was evaluated as PrEP in both vaginal and rectal SHIV challenge models. Although complete protection was achieved, the ISL levels were well above the limit deemed to be safe in humans [46]. Next-generation PCL implants releasing ISL at levels that are predicted to be safe were assessed as PrEP in the vaginal SHIV challenge model, and the threshold of protection is very close to the human PrEP benchmark of 50 fmol/10^6^ PBMCs [48]. Defining the threshold of protection for various routes of infection and confirming safe and sustained release are priorities for ISL implant modalities.

Although we document the sustained release of ISL from PCL implants, two out of six ISL-98 mg implants broke approximately 2 months after insertion. Implant breakage was associated with a concurrent spike in plasma ISL and ISL-TP in PBMCs, which led to redness and swelling at the implant site, albeit not as severe as intact TAF implants [24]. Increased levels of ISL and ISL-TP were maintained for over a month before the implants were removed, indicating that the drug remained in the implant despite the PCL wall being compromised. The cause of the implant breakages cannot be determined with certainty but may be related to several factors including the wall thickness, length, or a combination of the wall thickness and length of the implant. It is important to note that the ISL-62 mg implants, which had the same wall thickness (100 µm) as the ISL-98 implants but were shorter in length, all remained intact. Additional studies are underway to increase the mechanical robustness of PCL implants to ensure implants are capable of long-term safe and effective drug release [48].

Biodegradable PCL implants releasing ENG were also assessed to determine the compatibility of ISL and ENG as an MPT. ENG-33 mg implants placed in macaques achieved similar plasma ENG pharmacokinetics to those observed in humans implanted with Nexplanon implants [10]. ENG prevents pregnancy by inhibiting ovulation, which precludes the development of the corpus luteum, in turn limiting the production of endogenous progesterone [49]. To assess the efficacy of ENG, we used pigtailed macaques, which, like humans, have a lunar menstrual cycle and associated monthly hormone fluctuations. We measured progesterone as a surrogate of ovulation and found that progesterone production was suppressed until ENG implant removal, after which the progesterone levels rebounded, indicating a return to fertility [50]. 

Based on the different metabolic pathways of ISL and ENG, we did not expect significant drug interactions. Islatravir does not induce or inhibit major drug-metabolizing enzymes, including cytochrome50 (CYP), which is likely involved in ENG metabolism and has previously shown no DDIs with another progestogen contraceptive, levonorgestrel [51,52]. Consistent with this expectation, we found that co-administration of ISL and ENG resulted in no significant change to the PK profile of either drug. We found that ENG released in plasma had no effect on the pharmacokinetics of ISL in plasma or ISL-TP in PBMCs. However, we noted an unexpected decrease in dATP in vaginal tissues in the presence of ENG, which increased ISL-TP:dATP ratios. As dATP and ISL-TP compete for the active site of reverse transcriptase, a higher ratio of ISL-TP:dATP indicates a higher likelihood of ISL-TP incorporation and viral inhibition. Other studies have noted compartment-specific effects of hormonal treatments on nucleotides, both endogenous and antiretrovirals [39,40,41]. Further studies are needed to define the vaginal PrEP efficacy of ISL and possible modulations of activity in the presence of ENG or other hormonal contraceptives. 

This study assessed two ISL implants at three dose levels. However, the study had several limitations including a small sample size (*n* = 3 animals per dose) and a relatively short duration of 3 months, both of which hindered our ability to precisely determine the in vivo release rates and predict the duration of release. The accuracy of estimating in vivo release rates relies on achieving a zero-order release profile over the duration of the study. Additional investigations are required to further characterize the release profile, especially during the depletion phase. As the implant nears depletion, physiochemical differences could lead to altered release rates. Pharmacologic models, such as deconvolution, can estimate release rates based on the PK profile of ISL and that of the controlled-release formulation. Thus, these modeling techniques are not reliant on achieving a zero-order profile over the duration of the study to accurately estimate the release rate. A larger sample size and an extended study period will be important to determine drug pharmacokinetics through implant depletion, as well as inform on long-term safety, particularly regarding ISL exposure and potential effects on lymphocytes.

## 5. Conclusions

In summary, we describe a PCL implant with viable MPT attributes that can release ISL above predicted PrEP protection benchmarks and deliver ENG at levels that are sufficient to suppress ovulation for several months. The evaluation of optimized ISL-eluting implants with increased wall thickness is underway to assess long-term vaginal PrEP efficacy in a validated SHIV challenge model.

## Figures and Tables

**Figure 1 pharmaceutics-15-02676-f001:**
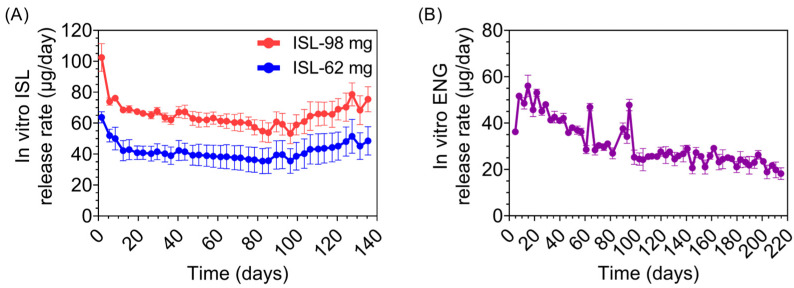
In vitro drug release from PCL implants. (**A**) Daily in vitro release profiles of ISL from implants run in parallel with the in vivo study in macaques with average release rates of 41.9 ± 5.5 µg/day (ISL-62 mg) and 65.6 ± 8.2 µg/day (ISL-98 mg). (**B**) Daily in vitro release profile of ENG from implants run in parallel with the in vivo study with an average release rate of 31.1 ± 9.6 µg/day.

**Figure 2 pharmaceutics-15-02676-f002:**
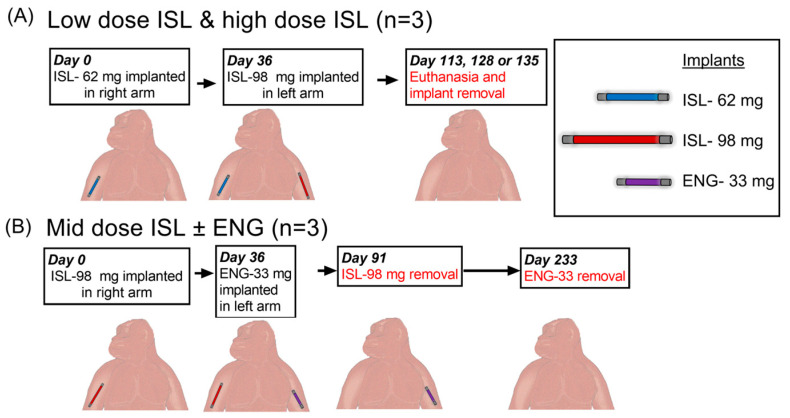
Study Design. (**A**) The first group of 3 macaques received ISL-62 mg implant (right arm) to assess the low ISL dose and a month later received an ISL-98 mg implant in their left arm to evaluate the high ISL dose. (**B**) The second group of 3 macaques received ISL-98 mg implant (right arm) to assess the mid ISL dose and received an ENG-33 mg implant in their left arm a month later. The ISL-98 mg implant was removed at 3 months. The ENG implant was removed after 6 months.

**Figure 3 pharmaceutics-15-02676-f003:**
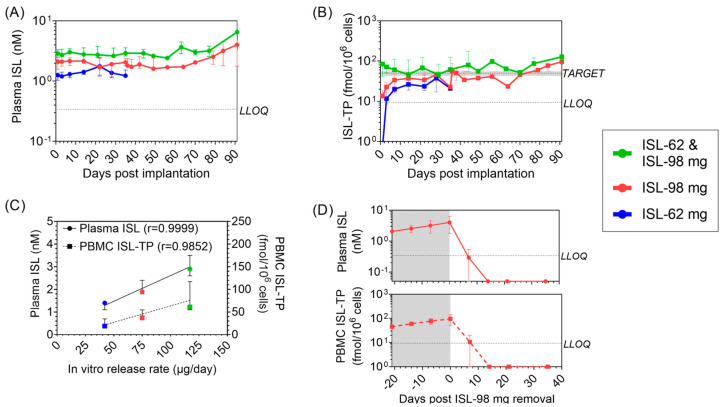
Pharmacokinetics of ISL and ISL-TP. (**A**) Median [range] plasma ISL and (**B**) PBMC ISL-TP levels for the low-dose ISL-62 mg (blue), mid-dose ISL-98 mg (red), and high-dose ISL-62 and ISL-98 mg (green). Day zero for ISL-62 mg and ISL-98 mg animals was the day of the first implantation. Day zero for the ISL-62 & ISL-98 mg was the day of the second implantation. The pharmacokinetic benchmark of 50 fmol ISL-TP/10^6^ PBMCs (Target) and lower limit of quantification (LLOQ) are shown with dashed lines. (**C**) Dose linearity between the in vitro release rate and plasma ISL (left axis, circles) and PBMC ISL-TP (right axis, squares). (**D**) Pharmacokinetic tail of ISL plasma ISL (top) and PBMC ISL-TP (bottom) after removal of ISL-98 mg from the mid-dose group.

**Figure 4 pharmaceutics-15-02676-f004:**
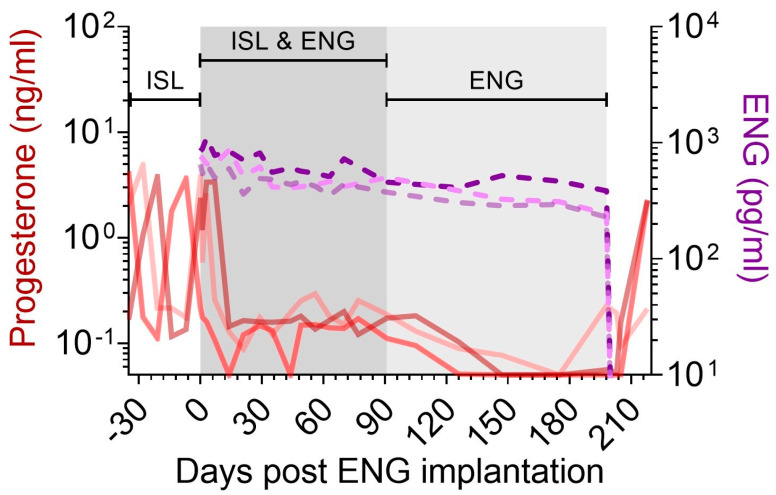
Plasma progesterone (left axis, red) and ENG (right axis, purple) in macaques that received ISL-98 mg and ENG-33 mg implants.

**Figure 5 pharmaceutics-15-02676-f005:**
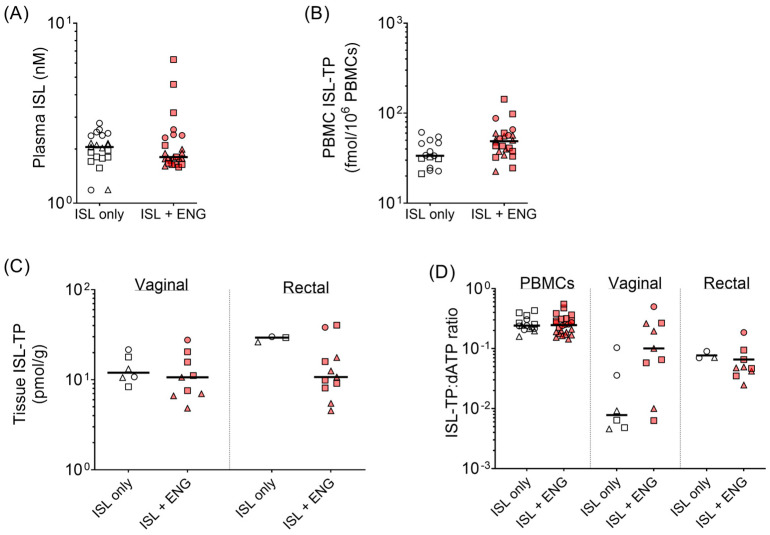
The effect of ISL and ENG co-implantation on ISL pharmacokinetics was determined by evaluating ISL analytes. (**A**) Plasma ISL, (**B**) PBMC ISL-TP, and (**C**) mucosal tissue ISL-TP when animals had a single ISL-98 mg implant (open shapes) compared to after co-implantation with ENG (red shapes). (**D**) Ratios of ISL-TP to the natural nucleotide competitor, dATP, were determined for each tissue. Each individual animal (*n* = 3) is shown with a different shape.

**Figure 6 pharmaceutics-15-02676-f006:**
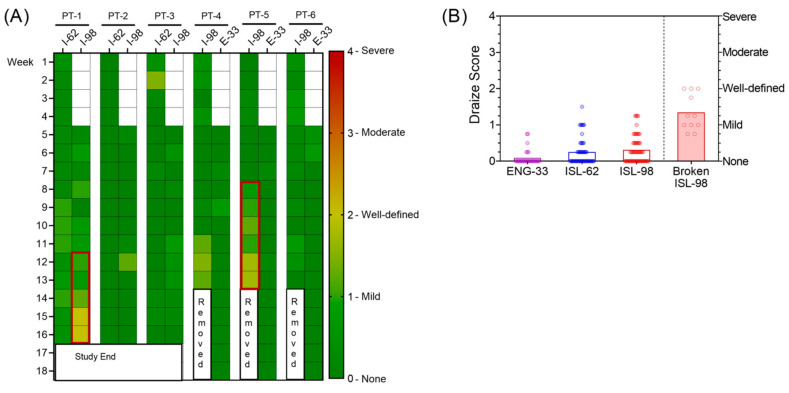
Implant site reactions were scored weekly for erythema and edema using a modified Draize scale. (**A**) Heat map of weekly Draize scoring of implant site for each animal (columns PT1-PT6) and each implant (ISL-62, ISL-98 or ENG-33 mg). Colors indicate severity of implant site reaction (0 = none/green, 1 = mid, 2 = well-defined, 3 = moderate, 4 = severe/red). Red boxes indicate when ISL-98 mg implants were deemed broken. (**B**) Median Draize scores for ENG-33 (purple), ISL-62 (blue), intact ISL-98 mg implants (red, unfilled box), and broken ISL-98 mg implants (red, filled box).

**Table 1 pharmaceutics-15-02676-t001:** Biodegradable ε-Polycaprolactone Implant Characteristics.

Implant	Length (mm)	Wall Thickness (µm)	In Vitro Release Rate (µg/Day)	In Vitro Purity (%)	In Vivo Purity (%)
ISL-62 mg ^1^	25	100	41.9 ± 5.5	97.1 ± 0.04	97.1 ± 0.09
ISL-98 mg ^1^	40	100	65.6 ± 8.2	97.1 ± 0.02	97.1 ± 0.12
ENG-33 mg ^2^	16	300	31.1 ± 9.6	99.3 ± 0.01	98.9 ± 0.01

^1^ ISL implants were formulated 1:1 with a sesame oil excipient. Average release rate ± standard deviation was determined from day 0–135. ^2^ ENG-33 mg implants were formulated 2:1 with a sesame oil excipient. Average release rate ± standard deviation was determined from day 0–215.

**Table 2 pharmaceutics-15-02676-t002:** Pharmacokinetics of ISL and ENG implants.

	ISL-62 mg (*n* = 3)	ISL-98 mg ± ENG-33 (*n* = 3)	ISL-62 mg + ISL-98 mg (*n* = 3)
Plasma			
Islatravir (nM)	1.3 [1.0–2.5]	1.9 [1.2–6.3]	2.8 [2.3–11.6]
ISL C_max_ (nM)	1.9 [1.5–2.5]	3.7 [2.2–6.3]	6.1 [2.9–11.6]
ISL T_max_ (days)	26 [22–35]	37 [7–91]	43 [7–91]
Etonogestrel (pg/mL)	n/a	495 [229–1100]	n/a
ENG C_max_ (nM)	n/a	875 [660–1110]	n/a
ENG T_max_ (days)	n/a	12 [7–22]	n/a
Peripheral blood mononuclear cells			
Islatravir-triphosphate (fmol/10^6^ cells)	20.5 [7.8–45.4]	37.7 [13.3–143.0]	63.6 [40.3 -229.0]
ISL-TP C_max_ (fmol/10^6^ cells)	36.4 [26.3–45.4]	96.8 [59.7–143.0]	164.2 [89.5–229.0]
ISL-TP T_max_ (days)	21 [14–28]	69 [36–91]	67 [44–100]
Mucosal tissues			
Vaginal ISL-TP (pmol/g)	9.6 ^a^ [8.5–11.6]	13.5 ^a^ [8.4–21.6]	30.5 ^b^ [10.7–64.1]
Rectal ISL-TP (pmol/g)	18.5 ^a^ [8.6–27.3]	20.5 ^a^ [12.5–34.0]	36.9 ^b^ [26.0–45.4]

Median [range] is shown for ISL and ENG in plasma and ISL-TP in PBMCs. Averages [range] are shown for C_max_, T_max_, and mucosal tissues. Data from broken implants was excluded. ^a^ Tissue concentrations 21 days after the first implant, either ISL-62 mg or ISL-98 mg, was inserted. ^b^ Tissue concentrations 21 days after the second implant, ISL-98 mg, was inserted.

**Table 3 pharmaceutics-15-02676-t003:** Implant site pathology.

Macaque ID	PT-4		PT-5		PT-6	
Biopsy Site	Medial	Distal	Medial (at Break)	Distal	Medial	Distal
Lymphocytes	1	0	3	1	0	0
Plasma cells	2	0	3	0	0	0
Histiocytes/Macrophages	1	0	3	0	0	0
Multinucleated giant cells	0	0	1	0	0	1
Polymorphonuclear cells	1	0	3	0	0	0
Fibrosis	0	0	3	0	0	0
Reactive fibroblasts	1	0	3	0	0	0
Hemorrhage	0	0	2	0	0	0
Neovascularization	0	0	2	0	0	0
Necrosis + type	0	0	2	0	0	0
Edema	0	0	3	0	0	0
Total Score	6	0	27	1	0	1

Total possible score: 55. Scoring criteria 0 = none; 1 = rare/minimal; 2 = mild/scattered; 3 = moderate; 4 = moderate to marked; 5 = abundant, extensive, severe. Scores recorded for lymphocytes, plasma cells, histiocytes/macrophages, polymorphonuclear cells, fibrosis, proliferating fibroblasts, and acute hemorrhage in the subcutaneous tissue (0 to 5 scale). Extravascular red blood cells only scored as hemorrhage if they were present deeper within tissue (not just at margins). Superficially adhered red blood cells consistent with acute hemorrhage at the time of surgical excision were not scored.

## Data Availability

The data presented in this study are available upon request from the corresponding author.

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
