# Peer review of "Pharmacokinetic Study of Islatravir and Etonogestrel Implants in Macaques"

_pharmaceutics, 2023, doi:10.3390/pharmaceutics15122676_

Round 1

Reviewer 1 Report

Comments and Suggestions for Authors

The manuscript describes a real-world macaque model for prevention of HIV infection and pregnancy by a reverse transcriptase inhibitor, islatravir and well established etonogesterel. The authors revealed that both implants little affect each other. Data presented are somewhat expectable and not surprising but provide some worth for the combination.

Data are not well organized nor discussed, and the present form of the manuscript seems in preliminary. For example, Fig 3C and D, and Fig 5D lack unit description.

1.      More detailed description or discussion should be added why authors chose ISL and 62 and 98 mg, and combination of 62 and 98 for this study. Moreover, ISL 98 mg was added on day 36. This is also difficult to understand.

2.      In the Figure 1, why release rate of both ISL groups was rapidly decreased and gradually increased, in early and late periods, respectively.

3.      The reviewer strongly recommends that the Figure 2 should be mentioned at the first.

4.      Addition of 98 mg implant how ISL and ISL-TP concentrations were changed?  

5.      How less than LLOQ could be determined?    

6.      In the Fig S1 B and C, increase of ISL was expectable after broken, however, why it is remained high level during the study?

Comments on the Quality of English Language

None

Author Response

Summary: The manuscript describes a real-world macaque model for prevention of HIV infection and pregnancy by a reverse transcriptase inhibitor, islatravir and well established etonogesterel. The authors revealed that both implants little affect each other. Data presented are somewhat expectable and not surprising but provide some worth for the combination.

We thank the reviewer for his/her comments which we feel improve the manuscript. Please see the corresponding revisions in the re-submitted file.

Comment 1: Data are not well organized nor discussed, and the present form of the manuscript seems in preliminary. For example, Fig 3C and D, and Fig 5D lack unit description.

Response 1: Units have been added to Fig 3C and D (same units as 3A and 3B). Fig 5D has been updated to ‘ISL-TP:dATP ratio’- there are no descriptive units.

Comment 2: More detailed description or discussion should be added why authors chose ISL and 62 and 98 mg, and combination of 62 and 98 for this study. Moreover, ISL 98 mg was added on day 36. This is also difficult to understand.

Response 2: This was a dose-ranging pilot study using i) ISL- 62 mg, ii) ISL-98 mg, and iii) ISL-62 mg and ISL-98 mg to understand the pharmacokinetics over a range of doses.

We have clarified the text in section 3.2 (pg.8, lines 323-327) and the legend of Figure 3 to better explain how the data is visualized in the subsequent PK figures. Animals that received ISL-62 mg implants received a second implant, ISL-98 mg, on day 36 after the first implant. To visualize all three doses concurrently, day zero for the high dose is the day of the second implantation. This staggered implantation strategy allowed for the testing of 3 doses with 3 animals per dosage group while using only 6 animals; outlined in Figure 2.

Comment 3: In the Figure 1, why release rate of both ISL groups was rapidly decreased and gradually increased, in early and late periods, respectively.

Response 3: The initial ‘burst’ release is common with certain configurations of implants until they achieve a steady state. The burst is dependent on multiple factors such as the implant length, wall thickness, drug, and excipients. Similarly, the rate of drug release may change over time due to features such as the depletion of drug and changes in molecular weight of the polymer. All in all, the release of drug from the implant maintained a sustained profile after the initial burst release throughout the 140 days of in vitro testing.

Comment 4: The reviewer strongly recommends that the Figure 2 should be mentioned at the first.

Response 4: We appreciate the reviewer’s feedback but anticipate that an earlier description of the in vitro release rates will help the reader to put the macaque data into the appropriate context, since these values are used to describe the macaque data.  We would thus prefer to keep the order as is.

Comment 5: Addition of 98 mg implant how ISL and ISL-TP concentrations were changed? 

Response 5: We clarify that the green lines in Figure 3 shows the data in the animals that had both the ISL-62 and the ISL-98 implants. To minimize confusion, we now indicate this in section 3.2 (pg.8, lines 323-327)) and the legend of Figure 3 to better explain how the data is visualized in the subsequent PK figures.

Comment 6: How less than LLOQ could be determined?   

Response 6: We have clarified in the Figure 3 legend that the median and range for each dosage group is being represented. In Figure 3D, we are showing the median (range) of the plasma ISL and PBMC ISL-TP for the mid-dose ISL-98 mg study group. Some animals had detectable drug, but others were below the LLOQ, and therefore range is both above below the LLOQ.

Comment 7: In the Fig S1 B and C, increase of ISL was expectable after broken, however, why it is remained high level during the study?

Response 7: The reviewer raises an important question. Unfortunately, we could not define with precision the day of implant breakage. We suspect that the levels remained high as the drug was still releasing from the compromised implant, i.e., there was likely still drug in the implant until removal and therefore drug levels remained high.

Reviewer 2 Report

Comments and Suggestions for Authors

The research wok “ Pharmacokinetic study of islatravir and etonogestrel implants in macaques” is preclinical experimental work on macaque model to evaluate the pharmacokinetics of biodegradable ε-polycaprolactone implants delivering the antiretroviral islatravir and the contraceptive etonogestrel effectively. The work is well design and novel. The comments are as follows:

1.       What is rational for combining this two drugs except patient compliance?

2.       Cite method is not developed in-house: 2.1 Implant fabrication and sterilization, 2.1 Implant fabrication and sterilization, 2.2 Determining release rates (in vitro) and drug purity (in vitro and in vivo), 2.3 Animal Care Guidelines, 2.4 Implantation and removal, 2.4 Implantation and removal, 2.6 Blood, tissue, and swab processing, 2.7 Measurement of intracellular ISL-TP and dATP from PBMCs and tissue, 2.9 Measurement of and progesterone in plasma and 2.10 Implant-site reactions

3.       2.2 Determining release rates (in vitro) and drug purity (in vitro and in vivo): HPLC Method (Provide as supplement file)?

4.       Supplement figure 1 and 2 is in the manuscript, please submit them separately.

Author Response

Summary: The research wok “ Pharmacokinetic study of islatravir and etonogestrel implants in macaques” is preclinical experimental work on macaque model to evaluate the pharmacokinetics of biodegradable ε-polycaprolactone implants delivering the antiretroviral islatravir and the contraceptive etonogestrel effectively. The work is well design and novel. The comments are as follows:

We thank the reviewer for his/her comments which we feel improve the manuscript. Please see the corresponding revisions in the re-submitted file.

Comment 1:       What is rational for combining this two drugs except patient compliance?

Response 1: We selected ENG since it is already FDA-approved as a long-acting contraceptive implant. ISL was chosen because of its potency and long intracellular half-life. The background for the selection of both drugs is discussed in the introduction.

Comment 2:       Cite method is not developed in-house: 2.1 Implant fabrication and sterilization, 2.1 Implant fabrication and sterilization, 2.2 Determining release rates (in vitro) and drug purity (in vitro and in vivo), 2.3 Animal Care Guidelines, 2.4 Implantation and removal, 2.4 Implantation and removal, 2.6 Blood, tissue, and swab processing, 2.7 Measurement of intracellular ISL-TP and dATP from PBMCs and tissue, 2.9 Measurement of and progesterone in plasma and 2.10 Implant-site reactions

Response 2: We are not sure we fully understand this question. We have added references to 2.3 Animal Care Guidelines (36) and 2.2 determining release rate in vitro (35).We provided appropriate references to methods that have been previously described including implant fabrication (21,35), drug level measurements (35, 37) and Implant-site reactions (23,39). The following methods do not have references and they were written for the manuscript: 2.4 Implantation and removal, 2.6 Blood, tissue, and swab processing, 2.9 Measurement of and progesterone in plasma.

Comment 3:       2.2 Determining release rates (in vitro) and drug purity (in vitro and in vivo): HPLC Method (Provide as supplement file)?

Response 3: We have updated the methods for 2.2 and added a citation where it is described in more detail.

Comment 4:       Supplement figure 1 and 2 is in the manuscript, please submit them separately.

Response 4: As suggested, these have been removed and we will submit these figures separately.

Reviewer 3 Report

Comments and Suggestions for Authors

The manuscript “Pharmacokinetic study of islatravir and etonogestrel implants in macaques

Abstract: The authors should include PK data for both drugs as a result in the abstract

Methodology: The number of macaques is a concern, only 3. Was it a pilot study before performing a big study/selecting the final formulation?

Results: PK data should ideally be plotted in pg/ng/µg and ordinal number instead of the scientific scale. It can be more didactic to the readers in general.

Author Response

We thank the reviewer for their comments which we feel improve the manuscript. Please see the corresponding revisions in the re-submitted file.

Comment 1: Abstract: The authors should include PK data for both drugs as a result in the abstract

Response 1: As requested, we have added the PK for both drugs in the abstract.

Comment 2: Methodology: The number of macaques is a concern, only 3. Was it a pilot study before performing a big study/selecting the final formulation?

Response 2: Yes, this was a pilot study to determine the safety of ISL and ENG implants and to assess the pharmacokinetics of ISL-eluting implants over a range of doses and the manuscript on second generation implants is in progress (reference 49). We have added language in the introduction to indicate this is a pilot study to better inform readers of the reason behind the small study groups. We have also edited the last paragraph of the discussion to better clarify that this work is preliminary with much still to be done.

Comment 3: Results: PK data should ideally be plotted in pg/ng/µg and ordinal number instead of the scientific scale. It can be more didactic to the readers in general.

Response 3: Thank you for this comment. This issue was extensively discussed by the authors prior to submission. We decided to use this scale and units to maintain consistency with the way implant clinical trials reported ISL data (see for instance, references 15-16). This will allow readers who are interested in the translational potential of this work to directly compare our findings to the clinical trial results.

Reviewer 4 Report

Comments and Suggestions for Authors

The authors described a PK study of three biodegradable PCL implants: ISL-62 mg, ISL-98 mg, and ENG-33 mg utilizing a preclinical macaque model. The ultimate goal is to develop a long-acting biodegradable implant for dual delivery of ISL and ENG to prevent HIV and unintended pregnancy. The paper was well written except for a few minor issues. I recommend the manuscript be reconsidered for publication after major revision.

1.       The sources/vendors for the internal standards are not listed in section 2.

2.        It is unclear why the authors chose different PCL thicknesses for the ISL and ENG implants.

3.       The PCL used for the three implants was claimed to be biodegradable but by the end of the study, the implants were found to be intact except for 2/6 ISL-98 mg implants. How long does it take for the implants to be completely degraded? What’s the removable window? When an implant starts to degrade in the body, will the authors expect to see side reactions similar to a broken implant? Will these side reactions limit the usable duration of the implants? How about the PK? Will the PK be affected by implant degradation?

4.       2/6 PCL implants broke during the PK study, which is a high brokage rate. Did the authors perform in vitro mechanical testing prior to the PK study or observe brokage while conducting in vitro dissolution? What was the rational for continuing the PK study for those two macaques with broken implants?

5.       For the high ISL dose group (ISL-62 mg + ISL-98 mg implants), when was considered as day zero? Was it when ISL-62 mg implants were implanted or when the ISL-98 mg implants were added over a month later? It is not stated clearly either in the main text or figure captions.

6.       The low dose ISL-62 mg implant group did not achieve the PrEP target of 50 fmol ISL-TP/106 PBMCs. Do the authors expect the clinical trial NCT05115838, which will evaluate ISL implants at even lower dose (47-57 mg), to fail to achieve the efficacy target as well? Are the implants used in this macaque PK study and the clinical trial similar?

Author Response

The authors described a PK study of three biodegradable PCL implants: ISL-62 mg, ISL-98 mg, and ENG-33 mg utilizing a preclinical macaque model. The ultimate goal is to develop a long-acting biodegradable implant for dual delivery of ISL and ENG to prevent HIV and unintended pregnancy. The paper was well written except for a few minor issues. I recommend the manuscript be reconsidered for publication after major revision.

We thank the reviewer for their comments which we feel improve the manuscript. Please see the corresponding revisions in the re-submitted file.

Comment 1: The sources/vendors for the internal standards are not listed in section 2.

Response 1: The vendors for the internal standards have been added.

Comment 2: It is unclear why the authors chose different PCL thicknesses for the ISL and ENG implants.

Response 2: We thank the reviewer for raising this important question. The release rate of drugs from the reservoir-style implant can be tailored by the wall thickness of the PCL tube. In this case, the authors performed many development studies in vitro to ultimately select the wall thickness that achieved the targeted drug release for this study. We clarified this concept on page 3, line 112.

Comment 3: The PCL used for the three implants was claimed to be biodegradable but by the end of the study, the implants were found to be intact except for 2/6 ISL-98 mg implants. How long does it take for the implants to be completely degraded? What’s the removable window? When an implant starts to degrade in the body, will the authors expect to see side reactions similar to a broken implant? Will these side reactions limit the usable duration of the implants? How about the PK? Will the PK be affected by implant degradation?

Response 3: Thank you for this comment. Unfortunately, this was a pilot study, and we could not address all of these important questions. We indicate at the end of the discussion the need for these additional studies and how they will inform on time to depletion, implant degradation, and effects on drug PK/toxicity. We hope these studies will also shed light on the window of implant usability.

Comment 4: 2/6 PCL implants broke during the PK study, which is a high brokage rate. Did the authors perform in vitro mechanical testing prior to the PK study or observe brokage while conducting in vitro dissolution? What was the rational for continuing the PK study for those two macaques with broken implants?

Response 4: We did not ordinarily observe breakage of these implant formulations during the in vitro dissolution assays. Of note, the in vitro dissolution conditions are gentle and may not fully emulate the mechanical perturbations present in a subcutaneous environment in vivo.

We suspected that the implant might have broken by visual inspection but unfortunately, we could not confirm this observation until implant removal. Also, we did not have the PK data in real-time to corroborate implant breakage. Animals were closely monitored (two or more times a week) after the implant was visually compromised and the implant site reactions were graded using the Draize scale scoring. Animals with broken implants remained on study because they showed no clinical signs of pain or distress, and this allowed us to assess any adverse reactions that may happen in a ‘worst case scenario’. Using a Draize score (mild to severe), implant sites were closely monitored to ensure adverse reactions were not progressing to a point that would endanger animal welfare. Ensuring the safety of animals on study including the increased monitoring of animals for adverse reactions were all predetermined and approved under our IACUC protocol.

Comment 5: For the high ISL dose group (ISL-62 mg + ISL-98 mg implants), when was considered as day zero? Was it when ISL-62 mg implants were implanted or when the ISL-98 mg implants were added over a month later? It is not stated clearly either in the main text or figure captions.

Response 5: Thank you for pointing out how this could be confusing for readers. The text section 3.2 and legend of Figure 3 has been updated to improve clarity for the readers.

Comment 6: The low dose ISL-62 mg implant group did not achieve the PrEP target of 50 fmol ISL-TP/106 PBMCs. Do the authors expect the clinical trial NCT05115838, which will evaluate ISL implants at even lower dose (47-57 mg), to fail to achieve the efficacy target as well? Are the implants used in this macaque PK study and the clinical trial similar?

Response 6: This is a very insightful comment. This number (i.e., 62 mg) represents the initial quantity of drug load in the implant, which does not solely dictate the amount of drug released since drug release may be affected by other parameters including wall thickness, length, and drug/excipient combination. Unfortunately, the clinical trials have not published the ISL release rates nor the specifics of their implant design. Also, our implants are biodegradable which differs from their non-erodible product. Importantly, the Merck implants achieved and maintained PBMC ISL-TP levels above the efficacy target for 12 weeks as described in ref 15 and 16. We have presented preliminary vaginal efficacy data using implants that achieve >50 fmol ISL-TP /10^6 cells in macaques (reference 49), and suspect that similar levels of efficacy can be achieved in humans.

Round 2

Reviewer 1 Report

Comments and Suggestions for Authors

For the Commnt 7, the authors must be menttioned the observation either in the Results section or the Discussion section as a limitation.

Other parts are adequately improved.

Comment 7: In the Fig S1 B and C, increase of ISL was expectable after broken, however, why it is remained high level during the study?

Response 7: The reviewer raises an important question. Unfortunately, we could not define with precision the day of implant breakage. We suspect that the levels remained high as the drug was still releasing from the compromised implant, i.e., there was likely still drug in the implant until removal and therefore drug levels remained high.

Author Response

Reviewer 1- 2nd round

For the Commnt 7, the authors must be menttioned the observation either in the Results section or the Discussion section as a limitation. Other parts are adequately improved.

Response: We have added the observation and implications in the discussion section (pg 16,line 543-544)

Reviewer 1- 1st round

Comment 7: In the Fig S1 B and C, increase of ISL was expectable after broken, however, why it is remained high level during the study?

Response 7: The reviewer raises an important question. Unfortunately, we could not define with precision the day of implant breakage. We suspect that the levels remained high as the drug was still releasing from the compromised implant, i.e., there was likely still drug in the implant until removal and therefore drug levels remained high.

Reviewer 4 Report

Comments and Suggestions for Authors

The authors have adequately addressed my comments.

Author Response

We thank the reviewer for taking the time to review this work and improving the content through the peer review process.